Primary health clinic toilet/bathroom surface swab sampling can indicate community profile of sexually transmitted infections

Giffard Philip M. phil.giffard@menzies.edu.au 1 2
Su Jiunn-Yih 3 4
Andersson Patiyan 1
Holt Deborah C. 1
1 Division of Global and Tropical Health, Menzies School of Health Research, Charles Darwin University , Darwin , Northern Territory , Australia
2 School of Psychological and Clinical Sciences, Charles Darwin University , Darwin , Northern Territory , Australia
3 Division of Child Health, Menzies School of Health Research, Charles Darwin University , Darwin , Northern Territory , Australia
4 Centre for Disease Control, Northern Territory Department of Health , Darwin , Northern Territory , Australia
Smith Tara
Electronic publication date: 2017 Jun 22
Publication date: 2017
Volume: 5
Electronic Location ID: e3487
Received 2017 Mar 22; Accepted 2017 Jun 1
Copyright: ©2017 Giffard et al.
Copyright year: 2017
Copyright holder: Giffard et al.
License: This is an open access article distributed under the terms of the Creative Commons Attribution License, which permits unrestricted use, distribution, reproduction and adaptation in any medium and for any purpose provided that it is properly attributed. For attribution, the original author(s), title, publication source (PeerJ) and either DOI or URL of the article must be cited.
License URL: https://creativecommons.org/licenses/by/4.0/

Keywords: Sexually transmitted infections, Chlamydia, Surveillance, Built environment, Swab sampling, Northern Australia

Funding: Australian National Health and Medical Research Council 1060768 This study was funded by Australian National Health and Medical Research Council project grant 1060768. The funders had no role in study design, data collection and analysis, decision to publish, or preparation of the manuscript.

==============================
Background

The microbiome of built environment surfaces is impacted by the presence of humans. In this study, we tested the hypothesis that analysis of surface swabs from clinic toilet/bathroom yields results correlated with sexually transmitted infection (STI) notifications from corresponding human populations. We extended a previously reported study in which surfaces in toilet/bathroom facilities in primary health clinics in the Australian Northern Territory (NT) were swabbed then tested for nucleic acid from the STI agents Chlamydia trachomatis, Neisseria gonorrhoeae and Trichomonas vaginalis. This was in the context of assessing the potential for such nucleic acid to contaminate specimens collected in such facilities. STIs are notifiable in the NT, thus allowing comparison of swab and notification data.

Methods

An assumption in the design was that while absolute built environment loads of STI nucleic acids will be a function of patient traffic density and facility cleaning protocols, the relative loads of STI nucleic acids from different species will be largely unaffected by these processes. Another assumption was that the proportion of swabs testing positive for STIs provides a measure of surface contamination. Accordingly, “STI profiles” were calculated. These were the proportions that each of the three STIs of interest contributed to the summed STI positive swabs or notifications. Three comparisons were performed, using swab data from clinics in remote Indigenous communities, clinics in small-medium towns, and a single urban sexual health clinic. These data were compared with time and place-matched STI notifications.

Results

There were significant correlations between swab and notifications data for the both the remote Indigenous and regional data. For the remote Indigenous clinics the p values ranged from 0.041 to 0.0089, depending on data transformation and p value inference method. Further, the swab data appeared to strongly indicate known higher relative prevalence of gonorrhoeae in central Australia than in northern Australia. Similarly, the regional clinics yielded p values from 0.0088–0.0022. In contrast, swab and notifications data from the sexual health clinic were not correlated.

Discussion

Strong correlations between swab and notifications were observed. However, there was evidence for limitations of this approach. Despite the correlation observed with the regional clinics data, one clinic yielded zero positive swabs for C. trachomatis, although this STI constituted 25.1% of the corresponding notifications. This could be ascribed to stochastic effects. The lack of correlation observed for sexual health clinic data was also likely due to stochastic effects. It was concluded that toilet/bathroom surface swab sampling has considerable potential for public health surveillance. The approach may be applicable in situations other than primary health clinics, and for targets other than STIs.

Introduction

Recent microbiome and metagenome-based analyses support the notion that the surface microbiomes of human inhabited built environments are closely connected to the human microbiome (Afshinnekoo et al., 2015; Flores et al., 2011; Hsu et al., 2016; Lax et al., 2014; Meadow et al., 2014; Schaumburg et al., 2016). This raises the possibility that microbiological analysis of built environment surfaces could provide information of public health utility; for example the surface may serve as a sentinel for microbial species, strains, genes or gene variants of interest. The advantages of such an approach to surveillance are that it is potentially simple and cost effective, and the specimens can be obtained without any burden upon or identification of individual patients or research participants.

Recently, a longitudinal multi-site assessment of sexually transmitted infection (STI) agent nucleic acid contamination of surfaces in clinic toilet/bathroom facilities in the Australian Northern Territory (NT) was reported (Andersson et al., 2014). The aim was estimation of the potential for such contamination to lead to positive STI tests performed on urine specimens, so the study also included simulation of urine specimen collection using a synthetic urine substitute. The STIs targeted were Chlamydia trachomatis, Neisseria gonorrhoeae and Trichomonas vaginalis. Surface contamination with STI nucleic acid was found in all clinics except for the sexual assault clinics, with the heaviest contamination in the remote primary health clinics. The frequency of simulated urine specimens testing positive for STIs was low but non-zero. Other studies in the UK have shown that STI agent nucleic acid contamination of clinic surfaces can readily be detected (Lewis et al., 2012; Meader, Waters & Sillis, 2008).

In the NT, STIs are notifiable, and notifications are placed in the public domain in the form of de-identified and aggregated data reported quarterly with some geographic breakdown (Northern Territory Department of Health). We reasoned that availability of time and locality matched toilet/bathroom surface swabbing data STI notifications data provided a unique and compelling opportunity to determine whether the results from toilet-bathroom swabbing are predictive of notifications. Surface cleaning protocols in clinic toilet/bathroom facilities will potentially be highly variable, and this would be expected to impact upon absolute STI nucleic acid load. Therefore, we did not regard it as useful to test the conjecture that absolute STI nucleic acid load on clinic toilet-bathroom surfaces predicts absolute STI prevalence in the community. Rather, the experimental design was based on the premise that while toilet/bathroom cleaning protocols would be likely to greatly impact absolute STI nucleic acid prevalence on toilet/bathroom surfaces, they would have much less impact on the relative loads of nucleic acid from different STI agents. Therefore, the conjecture we tested was that the relative prevalences of different STI agents in clinic toilet/bathroom swabs would be correlated with the relative prevalences of these STIs in the notifications data.

Methods

The basis of this study was the comparison of previously published data concerning STI agents in toilet/bathroom surface swabs (Andersson et al., 2014) with corresponding NT Government STI notifications data (Northern Territory Department of Health, 2017). The detailed and definitive procedure for the swab collection and analysis is described by Andersson and co-workers (2014). In summary, seven sites within the NT were included: two clinics associated with an NT Government organisation that investigates possible sexual assaults, two “sexual health” clinics in the Darwin and Alice Springs urban areas, two primary health clinics in small “regional” towns, and four primary health clinics in “remote Indigenous” communities. Each site was visited seven times over a period of approximately 12 months, and at each visit a standardised set of 10 toilet/bathroom surfaces were swabbed. The surfaces encompassed the floor, walls, toilet components, wash basin components and the door handle. At each visit, the experiment was performed in the male and female toilet/bathroom. The swabs were subjected to the same STI diagnostic procedure as actual clinical specimens for STI analysis from the clinic concerned i.e., the swabs were stored, transported to a diagnostic service provider, and subjected to C. trachomatis, N. gonorrhoeae, and T. vaginalis diagnostic procedures, along with actual STI diagnostic specimens generated on that day by the clinic concerned. Swabs from the sexual assault and sexual health clinic were analysed using the DNA-based Versant CT/GC DNA 1.0 Assay (kPCR) (Siemens, Munich, Germany), and swabs from the regional and remote clinics were analysed by a different service provider who used the RNA-based APTIMA Combo 2 Assay system (Gen-Probe, San Diego, CA, USA). The results of the swab analysis are presented in detail in Tables 1–3 of the paper by Andersson and co-workers (2014).

The current study encompassed swab data from the remote Indigenous and regional clinics, and the Darwin sexual health clinic. Two of the remote Indigenous clinics were located in the tropical “Top End” of the NT (close to the north coast), and swab data from these were compared with STI notifications data from the NT Government-defined “East Arnhem” region. The other two remote Indigenous clinics were located in Central Australia and the swab data were compared with “Alice Springs rural” STI notifications data. The numbers of positive swabs were normalised because of variation from the default number of 140 tests per STI agent per clinic. This variation was due to rare failure of service providers to complete tests. For example, for one remote clinic, 31 of 119 tested swabs were positive for T. vaginalis, and this was normalised to 35.3 positive swabs from 140 (Andersson et al., 2014). The swab and notifications data were compared on the basis of “STI profiles” comprised of the proportions contributed by each of C. trachomatis, N. gonorrhoeae and T. vaginalis positive tests/notifications to total positive swabs/notifications for these species.

The same approach was used to compare swab and notifications data from regional clinics. These are in towns that are not Indigenous communities, but have substantial Indigenous populations. In accordance with location, the swab data from one regional clinic was compared with notifications data from the Barkly region of the NT and the other compared with notifications data from the NT East Arnhem region. Finally, the same approach was used to compare swabbing and notifications data from a sexual health clinic located in Darwin.

The notifications data used were from 2012 (Northern Territory Department of Health, 2012a; Northern Territory Department of Health, 2012b). The great majority of the swabbing was performed in 2012, with a minority of the swabbing for a subset of clinics performed in late 2011 or early 2013 (Andersson et al., 2014). There were no major changes in STI relative prevalences in notifications data from late 2011 to early 2012, or from late 2012 to early 2013 (Northern Territory Department of Health), so we regarded comparing all swab data with 2012 notifications as justified. All the time and place specific notifications data were included in down-loadable documents in the public domain, (Northern Territory Department of Health, 2012a; Northern Territory Department of Health, 2012b) with the exception of the notifications data for the Darwin sexual health clinic for which a custom data extraction was performed.

Correlations were assessed using the Pearson Correlation Coefficient (r) for linear regressions of the swab and notifications proportions data. The logit transformation (log(x/(1−x))) and arcsine transformation (arcsin(x)) are reported to be useful in statistical analyses of proportions, with the logit regarded as superior, but not able to be used with proportions of 0 or 1 (Warton & Hui, 2011). In order to comprehensively explore the association between swab and notifications data, untransformed, logit transformed and arcsine transformed data were used. In addition, we minimised the requirement for assumptions regarding data distribution by inferring p values using a permutation approach. We developed macros in Microsoft Excel that for each data set performed 5,000 iterative randomisations of the order of one data column and recalculation of r. The inferred p value was the proportion of iterations for which r (randomised data) >r (non-randomised data, expressed with fewer decimal places than calculated values of r, and rounded down). p values were also calculated using the t statistic (Soper, 2017).

Ethical clearance for study was provided by the Human Research Ethics Committee of the Northern Territory Department of Health and the Menzies School of Health Research. The permit is HREC 2016-2696, “Environmental contamination as a predictor of community sexually transmitted infections profile.”

Figure 1 Correlations of STI profiles indicated by STI notifications and toilet/bathroom swab data.

The axes represent percent of total C. trachomatis + N. gonorrhoeae + T. vaginalis notifications/swabs. Where data points are labelled, the first or only letter indicates the organism, “C”: C. trachomatis; “G”: N. gonorrhoeae; “T”: T. vaginalis. The second pair of letters (where present) indicates location, “TE”: Top End”; “CE” Central Australia; “BA”: Barkly; “EA”: East Arnhem. (A) Remote Indigenous communities, (B) Small “regional” towns (swabs) and corresponding areas (notifications), (C) Darwin sexual health clinic, (D) Combined Remote and Regional data.

Results

First, we analysed data from 420 toilet/bathroom surface swabs from remote Indigenous communities. This experiment was of particular interest because these clinics generated the highest proportion and number of STI positive swabs, maximising statistical power. Also, remote primary health clinics are broadly similar in their characteristics and client base throughout the NT, but the notifications data indicated that the relative prevalence of N. gonorrhoeae was much higher in Alice Springs rural than in East Arnhem at the time the swabbing was performed, which we predicted would be reflected in swab data. From the tests for three STI agents on the 420 swabs, 208 positive results were obtained (Andersson et al., 2014). The STI profile defined by these results was compared with the breakdown of time and place-matched STI notifications as described in the Methods (Fig. 1, Table 1). There was statistical support for correlation of STI profiles. For all three transformations of the data, the permutation method yielded a higher (i.e., inferior) p value than the t-statistic method, but still indicated p values <0.05 (Table 1). The p values from the logit and arcsin transformed data were both lower (i.e., superior) than from the untransformed data (Table 1). The N. gonorrhoeae data were of particular interest because N. gonorrhoeae notifications are a much smaller proportion of the total in East Arnhem than in Alice Springs Rural (Northern Territory Department of Health, 2012a; Northern Territory Department of Health, 2012b). The correlation of this with the swabbing data is striking. N. gonorrhoeae contributes 4.6% of the positive swabs and 9.5% of the notifications from the Top End/East Arnhem, and 41.5% of the positive swabs and 46.1% of the notifications from Central Australia/Alice Springs Rural. Complete numeric data are provided as Supplemental Information 1.

Table 1 Correlation coefficients (r) and p values (p) for STI profiles derived from swab and notifications data.

The p values are single tailed. p(perm): p values derived from a permutation test. p(t): p values derived from t statistic.

Data transformation	untransformed	logit	arcsin	
clinics	r	p (perm)	p (t)	r	p (perm)	p (t)	r	p (perm)	p (t)	
Remote	0.857	0.041	0.015	0.891	0.015	0.0086	0.869	0.019	0.012	
Regional	0.945	0.0088	0.0022	NDa	NDa	NDa	0.924	0.0048	0.0042	
Both	0.898	<2.0 × 10−4	3.66 × 10−5	0.887b	0.0024b	0.0016b	0.870	<2.0 × 10−4	1.17 × 10−4	
Notes.

a “Not Done”. These were not done because four of the six points from regional clinics encompassed either 0% or 100% proportions in the swab data. The logit transformation cannot be performed on 0% or 100% values.

b Calculated from all remote clinic data and the two regional clinic points that do not encompass 0% or 100% proportions in the swab data.

For the regional clinics, 34 positive STI tests were derived from 280 swabs. The swab results from the two clinics were very similar to each other in that the great majority (30/34, 88%) of positive tests were for T. vaginalis. Despite the smaller number of positive tests, the correlation between swabs and notifications was stronger than for the remote clinics (Fig. 1, Table 1). Interestingly, the arcsin transformation improved the p value derived from the permutation method but had the opposite effect on the p value derived using the t statistic. However, as with the Remote Indigenous data, the p values from the permutation method indicated weaker significance than the p values derived using the t statistic.

For the Darwin sexual health clinic, the number of positive STI tests was lower still, with 10 positives from 140 swabs. Nine of the ten positive swabs were positive for N. gonorrhoeae, and with the remaining swab positive for T. vaginalis. This does not correlate with the notifications, for which the majority are C. trachomatis (Fig. 1). To explore possible reasons for the lack of correlation, we examined the raw data from the swabbing study (Andersson et al., 2014) and determined that five of the nine N. gonorrhoeae positive swabs were from a single toilet/bathroom (male), on a single day, and two of the remaining positive swabs also coincided in time and place of origin. Therefore, we can account for these nine N. gonorrhoeae positive swabs with just four surface contamination events.

We also analysed the data from the remote and regional clinics together. All p values obtained were ≤0.0024, and were less than 2 × 10−4 when all 12 points could be included (untransformed and arcsin transformed data) (Table 1), thus providing additional support that the correlation between the swab and notifications is significant (Fig. 1).

Discussion

This study illustrates both the potential value and some limitations of surface swabbing of clinic toilet/bathroom facilities for inferring STI profile in the community. We predicted that correlation would be highest with data from the remote clinics, because the large number of positive swabs (208) would minimise sampling error. Consistent with this, clear correlation was observed, and it appeared that the difference in relative prevalence of N. gonorrhoeae in remote central Australia and East Arnhem was reflected in the swab data.

Somewhat unexpectedly, the correlation from the “Regional” experiment was better than for the “Remote Indigenous” experiment, despite there being much fewer positive swabs. However, close examination of the data from the regional clinics does suggest that stochastic effects are evident. While the swab data are consistent with T. vaginalis having the highest relative prevalence in the relevant populations for both regional clinics, zero swabs from one of the clinics were positive for C. trachomatis or N. gonorrhoeae, despite these STIs representing 25.1% and 10.4% of the notifications in the region used for the comparison. Similarly for the other regional clinic, zero swabs were positive for N. gonorrhoeae, despite this representing 24.9% of the notifications used for the comparison. Thus the swab data from the regional clinics has failed to indicate the presence of STIs that comprise sizeable proportions of the notifications data.

The swab data from the sexual health clinic failed to reveal any useful information regarding the relative prevalances of STIs diagnosed in the same clinic. A clear anomaly is the lack of swabs positive for C. trachomatis, despite these constituting 79.2% of the notifications. The basis for this is unknown, but we speculate it is related to frequent cleaning of toilet bathrooms in this facility. This begs the question as to why N. gonorrhoeae positive swabs were obtained. As stated in the “Results”, five of the nine N. gonorrhoeae positive swabs arose from a single swabbing visit to a single male toilet/bathroom, highly suggestive of a single contamination event accounting for the majority of the N. gonorrhoeae positive swabs. It is also noteworthy that the simulated urine collection procedure at that time and place generated an N. gonorrhoeae positive specimen (Andersson et al., 2014). Positive results from the simulated urine collection experiments were very rare in that study as whole, so this is suggestive of a single very significant contamination event. It is of interest that the swab data from the Alice Springs sexual health clinic were very similar, with seven positive swabs for N. gonorrhoeae, and one positive swab for each of T. vaginalis and C. trachomatis. Because a DNA diagnostic technology was used for the sexual health clinic swabs and RNA based method for the other swabs encompassed by this study, we cannot rule out that STI diagnostic method affects the correlation between swab data and notifications data. While we regard this as unlikely, primarily because of the robustness and intensive quality control of commercial STI diagnostic systems, this may be an area worthy of future study.

We examined the data for other associations of interest. It was previously noted that positive swabs from female toilets are much more likely to be positive for T. vaginalis than swabs from male toilets, while the reverse is true for N. gonorrhoeae (Andersson et al., 2014). This is consistent with known disease processes, with T. vaginalis infection generally being regarded as more likely to cause symptoms and discharge in women than in men (Muzny & Schwebke, 2013), and N. gonorrhoeae being usually asymptomatic in women (Mayor, Roett & Uduhiri, 2012). Examination of the primary data from the surface swabbing study revealed that this bias is most extreme in the regional and sexual health clinics, with all 34 T. vaginalis positive swabs from regional clinics being from female toilets, and all 16 of the N. gonorrhoeae positive swabs from the sexual health clinics being from male toilets. It should be noted however that the single T. vaginalis swab from a sexual health clinic was from a male toilet/bathroom, and that association of N. gonorrhoeae with male toilets in the sexual health clinics may be related to the strong association of N. gonorrhoeae with men who have sex with men in the non-Indigenous population of Australia (The Kirby Institute, 2015). There were no N. gonorrhoeae positive swabs from regional clinics.

Our intention in conducting this study was not primarily to foreshadow implementation of toilet/bathroom surface swabbing for routine STI surveillance in the NT. There is no evidence that the STI notification system in the NT is ineffective. Indeed, it is the availability of comprehensive STI notification data that made it possible to investigate the informative power of the swabbing approach. We suggest that toilet/bathroom surface swabbing may have most potential utility in resource poor environments where there is no effective system of STI notification. However, we would not rule out toilet/bathroom swabbing could provide valuable information that is complementary to notifications data. A surface swabbing approach to STI surveillance encompasses people who have not been subjected to STI diagnostic procedures, and it may have potential to provide early indications of outbreaks. Possible applications of this approach may be in environments other than clinics, such as entertainment precincts or travel hubs. A built environment surface swabbing approach could also be very valuable in surveillance for targets other than STI agents, e.g., antibiotic resistance genes or exotic viruses.

In conclusion, the surface swabbing in the clinics in the remote communities in the NT was effective at indicating community STI profile, in large part because of the large number of STI positive swabs. It does appear, however, that different results will potentially be obtained from male and female toilets. While the expected association was also found for the “Regional” clinics, STI agents corresponding with STIs that formed an appreciable proportion of the notifications were not found in swabs. In the sexual health clinic, there was no correlation observed. We concluded that, as would be expected, this approach to surveillance can be confounded by low numbers of STI agent-positive swabs, leading to stochastic effects.

Supplemental Information

Supplemental Information 1 Complete numeric data.

Click here for additional data file.

The authors thank Dr. Peter Markey (Centre for Disease Control, Northern Territory Department of Health, Australia) for providing sexual health clinic specific notifications data, and Associate Professor David Whiley (Centre for Clinical Research, University of Queensland) for helpful discussions.

Additional Information and Declarations

Competing Interests

Author Contributions

Ethics

Data Availability

The authors declare there are no competing interests.

Philip M. Giffard conceived and designed the experiments, performed the experiments, analyzed the data, wrote the paper, prepared figures and/or tables, reviewed drafts of the paper.

Jiunn-Yih Su performed the experiments, analyzed the data, contributed reagents/materials/analysis tools, reviewed drafts of the paper.

Patiyan Andersson conceived and designed the experiments, performed the experiments, analyzed the data, reviewed drafts of the paper.

Deborah C. Holt conceived and designed the experiments, performed the experiments, analyzed the data, wrote the paper, reviewed drafts of the paper.

The following information was supplied relating to ethical approvals (i.e., approving body and any reference numbers):

The Human Research Ethics Committee of the Northern Territory Department of Health and the Menzies School of Health Research provided ethical clearance for this study (permit number 2016-2696).

The following information was supplied regarding data availability:

The raw data has been supplied as a Supplemental Information 1.

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
