# Peer review of "Primary health clinic toilet/bathroom surface swab sampling can indicate community profile of sexually transmitted infections"

_PeerJ, doi:10.7717/peerj.3487_

## Round 0.1 · original submission · Major Revisions

· Academic Editor

Major Revisions

Please pay careful attention to the reviewer comments, and especially the comments regarding clarity of methods.

Reviewer 1 ·

Basic reporting

While an interesting concept, there are several issues with this paper.

Reference to Carolyn Black's early work (decades ago) regarding amplicon contamination of STD clinic where CT NAATs are used would be an article that needs to be cited and any issues affecting this paper or supporting this paper's concept needs to be addressed.

The authors need to state key information from the Andersson et al. 2014 study in their paper so that readers do not have to return to the Andersson article for such information.

Experimental design

While there is reference to the Andersson paper, bathroom cleaning was not characterized or stated in the current paper itself by site, nor how cleaning was controlled.

This author's comment needs to be controlled for......"Rather, the experimental design was based on the premise that while toilet/bathroom cleaning protocols would be likely to greatly impact absolute STI nucleic acid prevalence on toilet/bathroom surfaces, they would have much less impact on the relative loads of nucleic acid from different STI agents."

The type of NAAT assay used e.g. TMA GenProbe , PCR Roche, SDA BD, etc.was not stated. Were different assays used at different sites.

No comment on RNA vs. DNA assays and their potential limitations in such sites as proposed.

The numbers results analyzed in some cases are too few. Further study is necessary to gain the conclusions the authors wish to present.

Validity of the findings

By the authors' own admission the concept fell down in the most likely clinic i.e. STD. Authors do note the issue, but go on to expel the result as ".....stochastic effects."
This is not sufficient. Since one would expect the STD clinic to have large numbers to add to the proposed premise, this need to be resolved fully

To avoid targeted prevalence studies by way of toilet swabbing for prevalence in remote or indigenous populations using such an approach as suggested may well do a disservice to those population as the paper does not sufficiently show correlation to supplant current public health methods.

Further study need to be accomplished before publication of these findings.

Reviewer 2 ·

Basic reporting

Review: Primary health clinic bathroom/toilet surface swab sampling…
Giffard et al.

The topic of this manuscript is of general interest to public health and, perhaps if it were executed properly, could be suitable for publication. Unfortunately, there are so many missing details and so much confusion in the writing that the paper needs a major revision. The discussion section is actually quite good and interesting, but the rest of the paper needs work.

The biggest problem in the paper, and the most serious one, is the utter lack of details regarding the methods ( I address this below) . There is so much missing information that it is nearly impossible to reconstruct what was actually done in the study.

The general point of the paper is also not clear. I assumed from the title that the authors were attempting to correlate detection of potential STI pathogens in the built environment (restrooms) with “notifications” (estimates if STI rates) in these communities in order to have an alternate biomarker for community STIs. This could potentially be useful for communities that did not have notifications of STI infections or for epidemiological studies. However, the authors do not clearly state the purpose or reason for the study in the abstract or introduction. What is the question/hypothesis? How will this help anyone? This should be front and center in the abstract and introduction. In general, I felt writing need to be overseen by someone with more microbiology and molecular biology experience. The use of the term “nucleic acid”, for example, is non-specific and inappropriate. Does the author mean DNA? RNA? Was it a PCR based assay? Was DNA extracted directly from the swabs? What it quantitated and if so how? So many unanswered questions.

I also thought that the figures were really sub-par. They graphs look like they came from an undergraduate lab notebook.

Experimental design

There were so many missing details in the methods that must be addressed. I realized the article cited another article, but that is just not good enough. Those details need to be in this paper.

The following must be addressed:
For the abstract and intro:
(1) What is the purpose of the study? What are its ramifications? What do the results tell you about the promise of the approach for public health?
For the methods in abstract and intro:
(2) Where exactly did the swabs come from? How were the surfaces swabbed? Who swabbed them and how much surface area was sampled? Was it the same on every restroom? I understand that the swab data were from a previous paper, but without more details on how the swabbing was completed, it is impossible to evaluate the approaches. I seemed to me like swabbing was done differently in all the restrooms by different people under different cleaning regimens. This all could really affect the results and maybe the fact that it didn’t always work. Details needed.
(3) How were the tests done? What at the details of the methods? Even brief details of the extraction process, and the tests that were completed would really improve the study. Was is qPCR, just pcr? Some other test? How good is this test?
For the results (abstract):
(4) For the correlations, the most important part is not the p-value, but the correlation value. The correlation values are extremely high for three of the four correlations in the figures (please add the p-value and r-values to the graphs). However, the abstract discussion indicates that the method didn’t work very well. I’m not sure why the authors don’t note the very strong correspondence, and focus only on the one health clinic with only 3 data points? I think the question is what is happening with that particular clinic?

For the discussion section in the abstract:
(5) The abstract is especially weak, but the authors need a question and a hypothesis, and the discussion should connect the two.

Validity of the findings

I think this could be a nice paper and the correlations are strong, but it is hard to evaluate with out more information and details on molecular methods and swab regimens.

Reviewer 3 ·

Basic reporting

Manuscript was well written, clear, unambiguous, professional English language used throughout.

Experimental design

Authors need to clarify the study compares data from a previous study (Anderson et al. 2014 to reported data NT.

Validity of the findings

Authors findings are noteworthy and reasonable.

Additional comments

This manuscript compares the relative prevalence of different STI agent in clinic toilet/bathroom swabs would be correlated with the relative prevalence of STIs in the notifications data in various regions in Australia.

In addition, the authors conducted the Pearson Correlation Coefficient (r) for linear regressions of the swab and notifications proportions data and conducted a t-test.

The authors conducted a well-organized study involving the comparison of surveillance data and reported (notification data). Overall, rationale of the study and methodological approach are reasonable. The authors’ major findings are scientifically sound and should be of interest to PeerJ readers. However, there are some specific comments that need to be addressed before it can be considered for publication.


Abstract:
Specific comments: (P: Page, L: Line)
The abstract is well written and clearly understood.

Introduction:
Specific comments:
L 62: Was the aim indicated here referring to Andersson et al. 2014 or this current study if so suggest authors indicate “The aim of this study was…”
L 66: “All specimens were analyzed …” This sentence seems this current study collected and analyzed these samples. Reference the study where the data was used from.
L 71: “Experimentation was…” do the authors mean samples were collected?
“transmission” … Change “transmission” to transmissible.


Methods:
Specific comments:
L 97: Do the authors mean in the present study? If so suggest to clarify this statement in order to be clear that the data from (Andersson et al. 2014) and the notifications data from NT was be compared. Suggest the authors clarify this throughout the methods

Results:
Specific comments:
L 144: Suggest authors clarify that the data analyzed 420 toilet/bathroom surface swabs ….was from (Andersson et al. 2014). Suggest to do the same throughout results.

Discussion:
Specific comments:
The discussion is well written.

---

## Round 0.2 · accepted · Accept

· Academic Editor

Accept

No additional changes are necessary. Thank you for your submission.